# Preparation of NH_4_Cl-Modified Carbon Materials via High-Temperature Calcination and Their Application in the Negative Electrode of Lead-Carbon Batteries

**DOI:** 10.3390/molecules28145618

**Published:** 2023-07-24

**Authors:** Meng Zhang, Hengshuai Song, Yujia Ma, Shaohua Yang, Fazhi Xie

**Affiliations:** 1School of Materials Science and Chemical Engineering, Anhui Jianzhu University, Hefei 230601, China; zm18240512051@stu.ahjzu.edu.cn (M.Z.); hssong0210@163.com (H.S.); wx1005080599@163.com (Y.M.); 2Anhui Accord Science and Technology Co., Ltd., Huangshan 242700, China; yangsh1985@163.com; 3School of Environment and Energy Engineering, Anhui Jianzhu University, Hefei 230601, China

**Keywords:** negative electrode additive, lead-carbon battery, hydrogen evolution reaction (HER), modified carbon materials

## Abstract

The performance of lead-acid batteries could be significantly increased by incorporating carbon materials into the negative electrodes. In this study, a modified carbon material developed via a simple high-temperature calcination method was employed as a negative electrode additive, and we have named it as follows: N-doped chitosan-derived carbon (NCC). The performance of this material was compared with a control battery containing activated carbon (AC). X-ray diffraction (XRD), scanning electron microscopy (SEM) and Raman spectroscopy were engaged in analyzing the crystal structure and morphology of the material. Afterwards, the electrochemical and battery performance was examined through cyclic voltammetry (CV), linear voltammetry (LSV) and constant current charge-discharge testing. Markedly, the electrode plate containing 1 wt.% NCC indicates the highest specific capacity (106.48 F g^−1^) as compared to the control battery, which is 1.56 times higher than the AC electrode plate and 4.75 times higher than the blank electrode plate. The linear voltammetry shows that the hydrogen precipitation current density of the 1 wt.% NCC electrode plate is only −0.028 A cm^−2^, a much higher value than that of the AC electrode plate. In addition, the simulated battery containing 1 wt.% NCC has a cycle life of 4324 cycles, which is 2.36 times longer than that of the same amount of additive AC battery (1834 cycles) and 5.34 times longer than that of the blank battery (809 cycles). In summary, NCC carbon has the advantage of extending the life of lead-acid batteries, rendering it a promising candidate for lead-acid battery additives.

## 1. Introduction

Lead-acid batteries have been widely applied in various areas for over a century, due to their low cost and superior discharge power, making it an important part of modern energy storage systems [1,2]. The sulfation of the negative active material (NAM) caused by the accumulation of PbSO_4_ in the high-rate partial-state-of-charge (HRPSoC) conditions is a main cause of battery failure, and lead-carbon batteries have emerged as a major solution to this problem. They have shown significant potential in the field of new energy storage and have been used to solve the problem of sulfation of the negative electrode [3]. The incorporation of carbon can enhance the reactive area of the NAM, and spatial resistance also inhibits the growth of PbSO_4_, thus reducing sulfation in lead-acid batteries [4].

In addition to suppressing sulfation in lead-acid batteries, carbon materials offer four other mechanisms of action, as described in [5]: Firstly, carbon materials can construct conductive networks, enhancing the electrical conductivity of the NAM. Secondly, carbon materials possess inherent bilayer capacitive properties. Thirdly, carbon can enhance the internal pore structure of the NAM in batteries. Finally, the addition of carbon materials to lead-acid batteries can promote superior electrochemical reaction dynamics, enhancing charging and discharging capacities. Boden et al. conducted a study on the effects of graphite, activated carbon (AC) and carbon black (CB) on lead-acid batteries and found that carbon materials can increase the electronic conductivity of the NAM and extend battery life [6]. Pavlov D. et al. investigated the charging and discharging processes of carbon and lead surfaces in lead-carbon batteries and proved that lead and carbon can form two parallel high-energy systems on the negative plate, resulting in a huge improvement in battery performance [7]. Shen et al. produced a bamboo leaf hierarchical porous carbon material (BLHPC) as an additive for the negative electrode of lead-carbon batteries. The high surface area and hierarchical porous structure of BLHPC improved the internal pore size of the NAM and promoted the penetration of electrolyte, and the team discovered that the battery life of BLHPC with a content of 0.5 wt.% increased 6.6 times compared to that of blank batteries [8]. Zhang et al. found that AC was served as an electron buffer during the charging process and exhibited better charge acceptance than blank electrodes in studying the electrochemical processes of AC in lead-carbon batteries [9]. However, carbon materials with low overpotential added to the negative electrode can cause serious hydrogen evolution reactions, making the electrolyte lose a lot of water, and destroying the structure of the lead-carbon batteries [10,11,12]. For example, Yin et al. found that the occurrence of HER disrupted the conductive network and affected cycling performance in their research on rice husk-derived hierarchical porous carbon batteries [5]. The current research focuses on the introduction of other metallic atoms into carbon materials to increase the material’s hydrogen evolution overpotential, while relatively less research has been carried out on the modification of carbon [13,14,15,16]. Hong et al. proposed that doping other atoms (N, P, B and S) in AC could effectively suppress HER and improve the battery’s performance [17]. Therefore, modified carbon materials have a greater prospect of usage in the field of energy storage.

In this paper, we prepared fluffy NCC materials through a simple high-temperature calcination process, characterized them via BET, XRD and SEM, and then we carried out electrochemical tests and battery tests as an additive in the negative electrode of lead-acid batteries. The results show that the NCC carbon materials have a large specific surface area and high porosity, and they are rich in heteroatoms. When used as an additive in the negative electrode of lead-acid batteries, the materials can effectively inhibit the growth of irreversible PbSO_4_, relieve the occurrence of HER to a certain extent, and extend the cycle life under HRPSoC conditions. The simulated battery with 1 wt.% NCC content contains a current density of −0.028 A cm^−2^ and a cycle life of 4324 cycles, which is 4.34 times higher relative to the blank battery (809 cycles). These results further support the effectiveness of NCC carbon materials as a potential additive for lead–acid batteries.

## 2. Results and Discussion

### 2.1. Characterization and Analysis of NCC Materials

The SEM images of the NCC material are shown in Figure 1a,b, which clearly show that the NCC material has a rich structure of bumps, grooves and corrugations, giving the material a dune-like appearance. During heating, spontaneous and stress-induced thermodynamic relaxation of the sugar precursors occurred. Additionally, impurities within the precursors lead to thermodynamic fluctuations that further contribute to the surface corrugations [18,19]. The presence of these corrugations give the NCC material a high specific surface area and greatly enhance the electrode plate electrochemical properties. The elemental mapping patterns of the NCC is shown in Figure 1c. The NCC material contains only three elements, namely, carbon (C), nitrogen (N) and oxygen (O), with atomic ratios of 77.09%, 8.04% and 14.87%, respectively.

The NCC materials were subjected to XRD tests to identify their crystal structure, as depicted in Figure 2a: The broad peaks observed at 23° and 43° in the NCC material indicate the amorphous structure of carbon, corresponding to the lattice planes (002) and (101) of graphite, respectively [20]. And, the sharper (002) peak of the NCC material indicates a higher degree of graphitization [21]. During the preparation process of NCC materials, chitosan treated with high-temperature heat gradually evolved into graphitized carbon, with a portion of the graphitic phase generated.

Raman spectra, an often-used method to investigate the degree of graphitization of carbon materials, were also deployed in this study, as shown in Figure 2b. The D-band at 1350 cm^−1^ represents the degree of disorder caused by the finite size of the crystal and defects, corresponding to the sp^3^ carbon domain. The G band at 1593 cm^−1^ is associated with C-C stretching in graphitic materials and corresponds to the sp^2^ carbon domain. The ratio of the D-peak to the G-peak intensity (I_D_/I_G_) can indicate the degree of disorder in carbon materials [22]. The I_D_/I_G_ value for NCC (0.42) is much lower than that of AC (0.99), implying a higher degree of graphitization of the NCC material. Resultantly, enhanced graphitization of carbon in the NAM corresponds to better electron conduction capacity and improved battery cycling efficiency.

The BET test was performed on NCC and AC materials, and the nitrogen adsorption curves at 77 K were employed to analyze the pore structure of the materials. The specific surface area of NCC is 404.36 m^2^ g^−1^, as shown in Figure 3a. The NCC exhibits a typical type-I isotherm, with a steepening at the low-pressure end (P/P^0^ = 0.2) and a bias towards the *Y*-axis. This indicates that NCC materials have a large number of micro- and mesoporous structures [23], which have a strong interaction with nitrogen. Meanwhile, its desorption curve overlaps with the adsorption curve without a hysteresis loop, highlighting the dominance of the microporous structure [24]. The pore size distribution was determined from their adsorption curves, and as shown, the pore size distribution peaks of the NCC samples ranged from 0 to 10 nm, confirming the microporous nature of the NCC material, which is consistent with the corresponding N_2_ adsorption and desorption curves. The specific surface area of the AC material is 806.60 m^2^ g^−1^, and its N_2_ adsorption and desorption curves exhibit type-II isotherms, reflecting the microporous characteristics of the material.

By performing XPS analysis of the material, we determined the surface chemistry of the NCC material, fitted the split peaks using the Thermo Avantage software (Avantage v5.9922), and we performed quantitative analysis. Figure 4 shows the XPS spectral analysis of the NCC composites. The full XPS spectrum (Figure 4a) shows the presence of only C, N and O elements in the sample with atomic ratios of 80.57%, 7.03% and 12.40%, respectively. The C 1s spectrum (Figure 4b) reviews the presence of two peaks, respectively, of the red curved peak at the binding energy of 284 eV, corresponding to C=N, and the blue curved peak at 285 eV, corresponding to C-O [25]. The N 1s spectrum (Figure 4c) exhibits a red curved peaking at 398.5 eV and a blue curved peaking at 402.5 eV, corresponding to -N= and graphitic N, respectively. The presence of these bonds implies the doping of N atoms in the NCC material. Furthermore, the incorporation of N in NCC materials enhances, which increases the cell capacitance in a certain way [26]. The O 1s spectrum (Figure 4d) displayed a red curved peak at 531 eV, corresponding to the presence of C-O. All green curves in the figure are baselines.

### 2.2. Electrochemical Properties of Electrode Plates

As displayed in Figure 5a, the electrochemical properties of the electrode plates were evaluated using a three-electrode system. The CV curves of the negative NCC plates with varying masses added are presented. The oxidation peak corresponds to the charging process, while the reduction peak corresponds to the discharging process. The oxidation and reduction peaks strengthen as the NCC content increases. The electrode plate with a 1 wt.% NCC addition has the most intense redox peaks. The high degree of overlap of the redox peaks suggests that the plate has good cycling performance. Figure 5b shows the CV diagram for the electrode plates containing 1 wt.% NCC, 1 wt.% AC, and a blank sample. By using the calculation Equation (1), the specific capacitance C (F g^−1^) of the electrode plates with different additives was obtained [27].
(1)$$C_{s}=\frac{\int i\left(V\right)d\left(V\right)}{2v\times ∆V\times m}$$
where $\int i\left(V\right)d\left(V\right)$ is the integrated area of the closed curve in the CV diagram, v is the scan rate (mV s^−1^), ΔV is the voltage range and m (g) is the mass of the carbon additive. The specific capacitance of the NCC electrode plates added at 0.5 wt.%, 1 wt.%, 1.5 wt.% and 3 wt.% is 19.03, 106.48, 62.22 and 24.71 F g^−1^, respectively; the specific capacitance of the 1 wt.% AC electrode plate is 68.35 F g^−1^; and the specific capacitance of the blank group electrode plate is 18.50 F g^−1^. Resultantly, the electrode plate containing 1 wt.% NCC has a much higher capacitance than other electrode plates, at 5.75 times greater than that of a blank electrode plate and 1.56 times greater than that of a 1 wt.% AC electrode plate. The results suggest that NCC electrode plates with a 1 wt.% NCC content provide the best electrochemical performance and exhibit the best electrochemical performance.

The LSV curves were obtained for electrode plates with different content of NCC, as shown in Figure 5c, and for electrode plates containing 1 wt.% NCC and 1 wt.% AC. And, the blank is shown in Figure 5d. The hydrogen precipitation current density of 0.5 wt.%, 1 wt.%, 1.5 wt.% and 3 wt.% NCC electrode plates at −1.1 V is 1 wt.% NCC > 1.5 wt.% NCC > 3 wt.% NCC > 0.5 wt.% NCC. Our findings suggest that the 1 wt.% NCC negative plate has the highest hydrogen precipitation current density of −0.028 A cm^−2^, effectively suppressing the HER. As shown in Figure 5d, the hydrogen evolution initiation potential magnitudes for the added 1 wt.% NCC, 1 wt.% AC and blank electrode plates are 1 wt.% NCC < blank < 1 wt.% AC. As the scanning voltage gradually moves to a lower voltage, the hydrogen precipitation current gradually decreases from a steady level and initiates the HER reaction at the electrode plate. Specifically, the 1 wt.% NCC plate has the lowest hydrogen evolution initiation potential and the latest onset of HER reaction compared to the other the electrode plates, so the HER €s effectively suppressed in the 1 wt.% NCC plate.

Lead–carbon batteries represent a new type of lead–acid battery; they exhibit a double layer of capacitive properties because of the presence of carbon. The addition of carbon prevents sulfation of the negative plates, resulting in a significant increase in the number of battery cycles.

In our study, we applied Nyquist diagrams obtained through a simulation of the equivalent circuit diagram of a lead–carbon battery, demonstrating the high-frequency circuit and the low-frequency curve, as shown in Figure 5e,f, respectively [28]. In the high-frequency region, we can see that the diameter of the semicircle first tends to decrease and then increase with the increasing amount of NCC added. This capacitive contribution relates to the contact between carbon atoms [29]. Remarkably, the smallest half-circle diameter of the 1 wt.% NCC electrode plate indicates the greatest increase in conductivity of the electrode plate with the addition of 1 wt.% NCC. In contrast, in the low-frequency region, the spectrum shows a clear linear behavior. The rate of diffusion of material transfer in the pores of the electrode plate is the determining factor in this region. Consequently, the difference in the angle of the impedance spectrum suggests the variation of pore structure parameters in these plates [30].

As data shown in Table 1, R_s_ is the ohmic resistance of the electrode plate, with a smaller value of R_s_ indicating a better conductivity of the electrode plate. The R_s_ value of the 1 wt.% NCC electrode plate is significantly smaller than that of the 1 wt.% AC electrode plate and the blank electrode plate, indicating that the 1 wt.% NCC electrode plate has the best conductivity. Furthermore, the Q_f_ and R_f_ values correspond to the non-ideal bilayer capacitance of the carbon material and the corrosion resistance of the NAM and Pb plate grids in the electrode plate, respectively [31]. The value of the charge transfer resistance R_ct_ is usually used to compare the kinetic activity, with the R_ct_ value being inversely proportional to the kinetic activity. As can be seen from Table 1, the R_ct_ values of the electrode plates in the NCC group were all smaller than those of the 1 wt.% CB electrode plates and the blank plates, indicating stronger kinetic activity. Q_dl_ represents the non-ideal electrical layer capacitance of the NAM [32,33], with the value proportional to the electrochemical effective area of the electrode plate. As shown in Table 1, the NCC gradually floats on the Pb surface as the NCC content increases, resulting in a gradual decrease in the Q_dl_ value of the electrode plate and a gradual decrease in the electrochemical effective area.

In order to explore the hydrogen evolution performance of NCC and AC materials, we analyzed the Tafel curves. As shown in Figure 5g, the absolute values of the Tafel slopes for the 0.5 wt.%, 1 wt.%, 1.5 wt.%, 3 wt.% NCC and 1 wt.% AC electrode plates are 0.2931 mV dec^−1^, 0.0717 mV dec^−1^, 0.1028 mV dec^−1^, 0.0510 mV dec^−1^ and 0.1387 mV dec^−1^, respectively. Markedly, the absolute value of the slope of the 1 wt.% NCC electrode plate is relatively small, indicating a significant suppression of HER. In summary, electrode plates containing 1 wt.% NCC are effective in boosting battery capacity while suppressing HER.

### 2.3. Battery Performance Test

Figure 6a shows the cycle life of all simulated batteries under HRPSoC operating conditions, and Table 2 shows the cycle life data for different additive groups of batteries. The test procedure involved charging for 30 s at 2C current, discharging for 30 s and then cycling the test until the discharge voltage reaches 1.75 V. The experimental results reveal that all carbon additives can improve the HRPSoC cycle life of lead–acid batteries. The battery voltage gradually decreases as the number of cycles increases, and the performance of the NCC group remains stable for the first 500 cycles with different levels of NCC; meanwhile, the performance of the batteries with 1 wt.% AC and the blank group is poor. The simulated battery with 1 wt.% NCC content at 2C current has the longest cycle life of 4324 cycles, which is 5.34 times that of a blank battery (809 cycles) and 2.36 times that of a 1 wt.% AC group battery (1834 cycles). These findings are consistent with the results of electrochemical tests. The addition of NCC improves the specific surface area and pore size of the NAM and increases the diffusion rate of HSO_4_^−^ in the electrolyte [33]. This suggests that NCC materials containing N elements may inhibit HER to some extent, thereby increasing the NAM utilization and extending battery life.

Figure 6b illustrates the discharge capacities of batteries containing different NCC contents, 1 wt.% AC batteries and blank batteries at different multipliers. As shown in the graph, the initial capacity of all lead–carbon batteries increases at a discharge rate of 0.1C, with the 1 wt.% NCC batteries displaying the highest discharge capacity. This is due to the improved negative conductivity of the simulated cell with the addition of the NCC material, leading to an increase in electrochemical reactivity [34]. However, at a discharge rate of 1C, the capacity of all batteries decreases. The results show that the cycle stability and discharge capacity of all lead–carbon batteries are superior to those of the blank batteries at different discharge multipliers. The batteries with 1 wt.% NCC exhibit the best cycling stability and discharge capacity, with the discharge capacity being significantly higher than that of other batteries, which is in line with its CV and LSV test results. The reduced and unstable nature of the discharge capacity of the blank battery when the discharge rate is again 0.1C is mainly due to the impact of the high current on the electrode plates destabilizing the battery structure after the high-current test. Furthermore, a high-current discharge leads to irreversible sulphation of the cell and reduced battery performance. 

The main reason for the better performance of NCC batteries In the tests may be that the NCC material has a large number of corrugated structures, providing a more specific surface area and pore size structure for the Pb/PbSO_4_ reaction interface and electrolyte transport, thus increasing the beneficial role of the carbon material in the negative electrode. In addition, the 1 wt.% NCC battery life of up to 4324 cycles may be due to the inhibitory effect of the N element contained in the NCC on HER, thereby reducing hydrogen precipitation on the surface of the battery and improving the utilization of the carbon material [17]. It also reduces the effect of the hydrogen spillage process on electrolyte transport in the material pore size and increases the Pb deposition rate. The drop in HER on the surface of the carbon material has the potential to further enhance the conductive network formed by the carbon material in the negative electrode and increase the electron transfer rate, which is conducive to increasing the conversion rate of PbSO_4_ to Pb.

## 3. Experimental Materials and Methods

### 3.1. Material Preparation

Chitosan and NH_4_Cl were purchased from Macklin Biochemical Co., Ltd (Shanghai, China). All reagents used in this study were analytically pure, and no further purification was required prior to use.

Pretreatment of chitosan: an appropriate amount of chitosan was soaked in an oxalic acid solution with a mass ratio of 2 wt.% for 1 h. The solution was then rinsed with ultrapure water until neutral, and the remaining powder was placed in a vacuum drying oven and dried at 40–50 °C for 12 h. The light-yellow powder was subsequently obtained by grinding.

Preparation of NCC materials: Firstly, the mixture of 10 g of pretreated sugar and NH_4_Cl was ground for 10 min and heated to 1000 °C in a N_2_ tube furnace and then held for 3 h until cooled to room temperature, and the black swelling was collected and ground into a black powder in an agate mortar, which is the NCC powder, as shown in Figure 7.

### 3.2. Preparation of Negative Plates

The NCC additive was mixed with the lead paste semi-finished product, along with a small amount of deionized water. The resultant mixture, known as mixture A, was obtained after further stirring in a paste mixing machine. The lead paste semi-finished product contained 0.1–0.15 wt.% lignin, 0.2–0.3 wt.% humic acid, 0.05–0.1 wt.% carbon fiber, 0.5–1.5 wt.% barium sulfate and 93–99 wt.% lead powder. Mixture A was subsequently evenly coated on the negative plate grid, flattened, acid-drenched, cured, and dried, until the negative plate was obtained.

### 3.3. Material Characterization and Electrochemical Testing

The NCC material was characterized and analyzed via X-ray diffraction (XRD, Bruker D8 Advance) and cold field emission scanning electron microscopy (SEM, Regulus 8100 Hitachi), and the nitrogen adsorption–desorption isotherm was measured using the Autosorb IQ (micromeritics ASAP 2460) instrument for the crystal structure of the material.

We prepared electrode plates with different contents of NCC additives (0.5 wt.%, 1 wt.%, 1.5 wt.% and 3 wt.%) as the experimental group, with 1 wt.% AC electrode plates as the control group, and no additive electrode plates as the blank group. All electrode plates were 1 × 2 cm in size, and the mass of NAM was 1300 mg. The analysis of the electrode plates was conducted using an electrochemical analyzer. A homemade electrode plate was employed as the working electrode, with a Pt electrode used as the counter electrode and the Hg/HgSO_4_ electrode as the reference electrode, and the electrolyte deployed was 1.23 g/mL of H_2_SO_4_ solution. The cyclic voltammetry (CV) was performed within a voltage range from −1.2 to 0.2 V with a sweep rate of 0.01 V/s and the linear scanning voltammetry (LSV) from −1.1 to 0.4 V with a sweep rate of 5 mV/s; the electrochemical impedance spectroscopy (EIS) was fitted with Zview.

The negative and positive plates were prepared with dimensions of 2 × 2 cm, and the respective NAM mass for the negative and positive plates was 2600 mg and 3800 mg. To assemble a closed lead–acid battery, two positive plates and one negative plate were separated using a 3 mm AGM spacer. The simulated battery was tested with the Newell Battery Test System (BTS Client 8.0.0.471, CT-4008Tn-5V6A-S1) for charge/discharge and multiplier testing.

## 4. Working Principle

To further investigate the mechanism of the lead–carbon battery, the negative plates were characterized by XRD before and after the HRPSoC cycle. Prior to the test, the simulated battery needed to be activated via a chemistry step. The XRD of the battery after chemistry is shown in Figure 8a. All six groups of batteries exhibit typical Pb and PbSO_4_ diffraction peaks. The intensity of the PbSO_4_ diffraction peak in the negative plate first decreases and then increases as the NCC content improves. The negative plate containing 1 wt.% NCC displays the fewest PbSO_4_ diffraction peaks, whereas the blank negative plate depicts a large number of PbSO_4_ diffraction peaks. The incorporation of carbon materials in the NAM can build a conductive network that facilitates the conversion of PbSO_4_ to Pb in the chemistry step, reducing the amount of PbSO_4_ in the negative plate and increasing the amount of active material Pb in the negative plate prior to cycling [34,35]. However, an excessive addition of NCC can lead to uneven mixing of Pb and C, resulting in an increase in the PbSO_4_ phase during the battery formation step. In summary, adding an appropriate amount of NCC material can improve the conversion of PbSO_4_ to Pb in the NAM, thereby changing the NAM composition, while excessive carbon incorporation will instead lead to accelerated battery failure.

Figure 8b shows the XRD diagram of the negative plate after the HRPSoC cycle test. We found that the negative plate is mainly composed of the PbSO_4_ phase. At the completion of the battery cycle, the intensity of the PbSO_4_ peak in the carbon negative plate is less than that of the blank negative plate, indicating that the addition of carbon material helps to reduce the accumulation of PbSO_4_ crystals in the negative plate. Remarkably, with the increase in the NCC content, the intensity of the PbSO_4_ diffraction peak first decreases and then increases, and the intensity of the PbSO_4_ diffraction peak of the negative plate containing 1 wt.% NCC is the weakest. The excessive addition of NCC leads to the carbon floating on the Pb surface, hindering the carbon material from suppressing the sulfation of the negative plate [36]. The intensity of the PbSO_4_ peak in the NAM of the 1 wt.% NCC battery is much weaker than that of the 1 wt.% AC negative plate. This is attributed to the large number of microporous and mesoporous structures in the NCC material, which can accelerate the rate of ion diffusion during the reaction and inhibit the aggregation and irreversible sulfation of PbSO_4_ crystals.

The SEM characterization of the HRPSoC was utilized before and after cycling, and the results are presented in Figure 9. And, Figure 9a–c are the microscopic morphologies of the blank battery, 1 wt.% AC battery and 1 wt.% NCC battery before cycling experiments, respectively. The NAM in the negative plate of the blank battery undergoes aggregation with a smooth surface containing large particulate matter, leading to a reduction in the active sites on the surface of the blank plate and a reduction in the charging efficiency of the battery [37]. In contrast, the morphology of the negative plate of the 1 wt.% AC battery and the 1 wt.% NCC battery is very different from that of the blank plate, with the presence of a large amount of rough and fluffy metallic Pb. This observation is consistent with the results in Figure 8a, which indicate that carbon can influence the microscopic morphology of the negative plate and plays an important role in the conversion of the NAM. As shown in Figure 9d–f, at the end of the cycling experiment, a large number of irreversible PbSO_4_ particles accumulated on the surface of the blank plate, hindering the diffusion of the electrolyte and ultimately resulting in a reduced battery life. Remarkably, the carbon-containing negative plates have significantly smaller PbSO_4_ particles compared to the blank plates, and the 1 wt.% NCC negative plates have the smallest PbSO_4_ particles. The addition of carbon material can effectively inhibit the growth of large PbSO_4_ particles on the surface of the negative plate, thus improving the cycle life under HRPSoC conditions. However, the negative plate containing 1 wt.% AC exhibits a shorter cycle life than a plate containing 1 wt.% NCC negative plates due to severe HER.

Following the analysis of the above experimental results, the working mechanism of NCC materials in the negative electrode plate of lead–acid batteries is shown in Figure 10 below. During the high-rate discharge process, the micropores and mesopores function as electrolyte transfer channels, facilitating the reaction of the electrolyte into the interior of the electrolyte. The larger specific surface area of the NCC material provides more reaction sites for the conversion of PbSO_4_ compared to blank plates, inhibiting the accumulation and growth of PbSO_4_ on the surface of the negative plates. As a result, NCC batteries display a higher battery performance under HRPSoC conditions compared to blank and AC batteries.

## 5. Conclusions

To summarize, we used chitosan as the carbon source, mixed it with NH_4_Cl in equal amounts and calcined it at 1000 °C using an N_2_ tube furnace to produce a fluffy and porous modified carbon material named NCC. The NCC material has a large number of micropores and mesopores and a high specific surface area, and it is rich in N atoms. The use of NCC as an additive to negative plates improves the utilization of the NAM. In electrochemical tests, the electrode plate containing 1 wt.% NCC exhibits the highest specific capacitance (106.48 F g^−1^), the highest current density for hydrogen precipitation (−0.028 g cm^−2^_)_ and the lowest resistance. And, in battery performance tests, the 1 wt.% NCC additive simulated battery has the highest cycle life (4324 cycles), at 5.34 times that of a blank battery, and 2.36 times that of the 1 wt.% AC battery. The raw material of the NCC is cheap, making it cost effective. Moreover, the addition of NCC has a low impact on the hydrogen precipitation of the electrode plate in electrochemical tests and can effectively improve the battery’s performance, so it is a promising material that can be used as a negative electrode additive in the battery industry on a large scale.

## Figures and Tables

**Figure 1 molecules-28-05618-f001:**
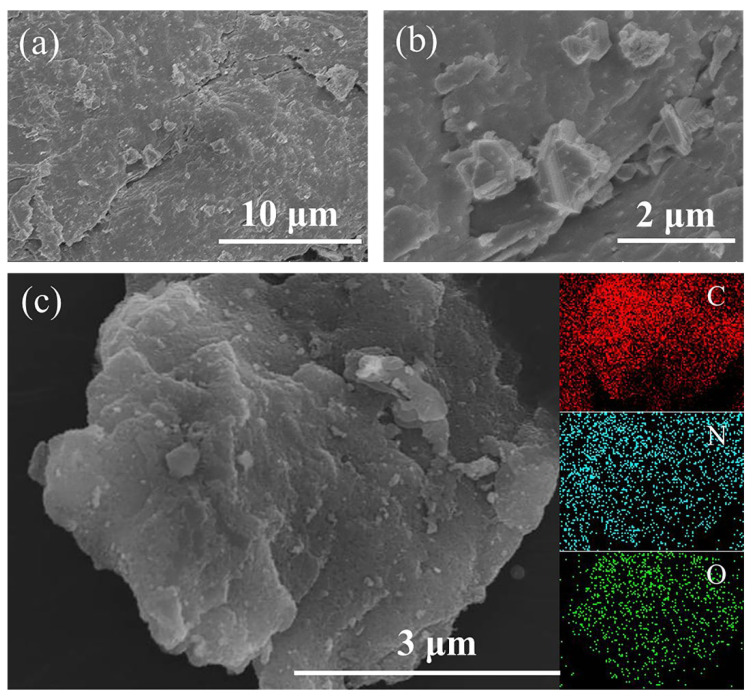
(**a**,**b**) SEM images of NCC materials at 10 μm and 2 μm; (**c**) elemental mapping patterns.

**Figure 2 molecules-28-05618-f002:**
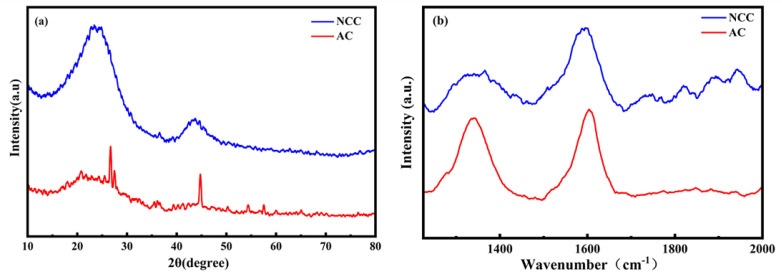
(**a**) XRD patterns of NCC and AC; (**b**) Raman spectra of NCC and AC.

**Figure 3 molecules-28-05618-f003:**
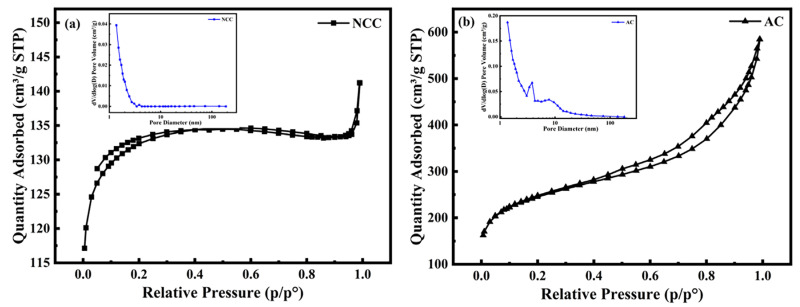
(a) Adsorption and desorption curves and pore size distribution of NCC; (b) adsorption and desorption curves and pore size distribution of AC.

**Figure 4 molecules-28-05618-f004:**
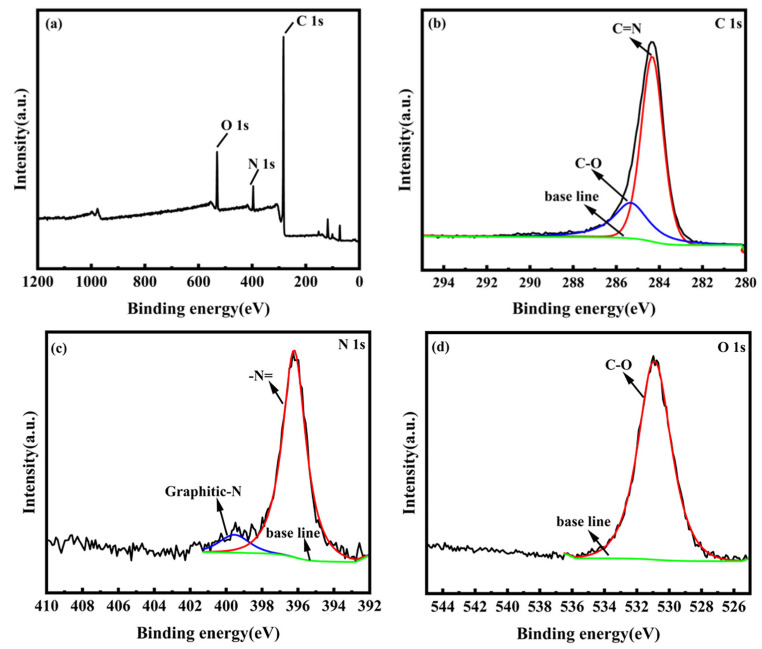
(**a**) Full spectrum of NCC composites; XPS spectra of (**b**) C 1s, (**c**) N 1s and (**d**) O 1s.

**Figure 5 molecules-28-05618-f005:**
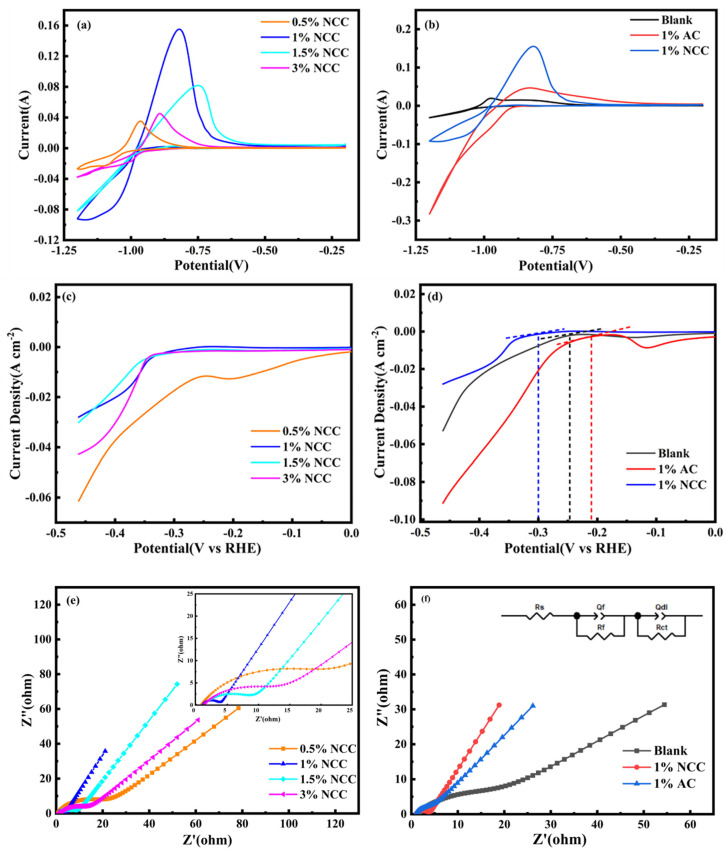
(**a**) CV diagram of NCC with different contents; (**b**) CV diagram of 1 wt.% AC, 1 wt.% NCC and blank group; (**c**) LSV diagram of NCC with different contents; (**d**) LSV diagram of 1 wt.% AC, 1 wt.% NCC and blank group; (**e**) EIS diagram of NCC with different contents; (**f**) EIS diagram of 1 wt.% AC, 1 wt.% NCC and blank group; (**g**) Tafel.

**Figure 6 molecules-28-05618-f006:**
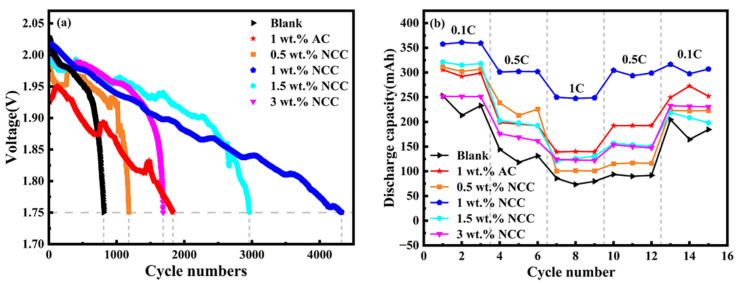
(**a**) At 2C current, the end-of-discharge voltage limits the battery life of HRPSoC; (**b**) discharge capacity of NCC, AC and blank group batteries at different multipliers.

**Figure 7 molecules-28-05618-f007:**
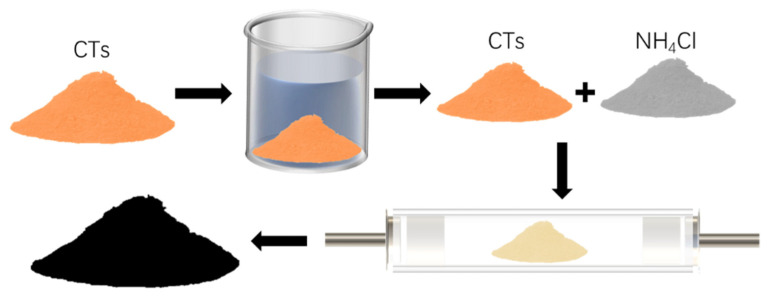
Production flow chart of NCC composite material.

**Figure 8 molecules-28-05618-f008:**
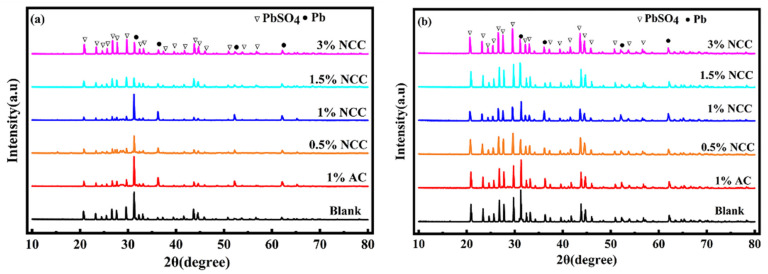
The XRD plots of different additive NAMs before (**a**) and after (**b**) the battery cycle test.

**Figure 9 molecules-28-05618-f009:**
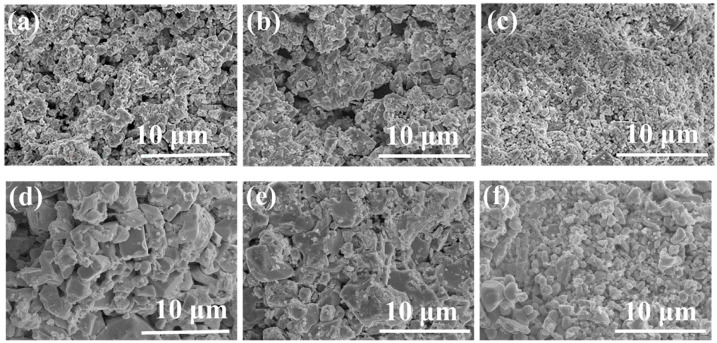
SEM images of the negative plate before (**a**–**c**) and after the HRPSoC cycle test (**d**–**f**): (**a**,**d**) blank negative plate; (**b**,**e**) 1% AC negative plate; (**c**,**f**) 1% NCC negative plate before and after the cycle.

**Figure 10 molecules-28-05618-f010:**
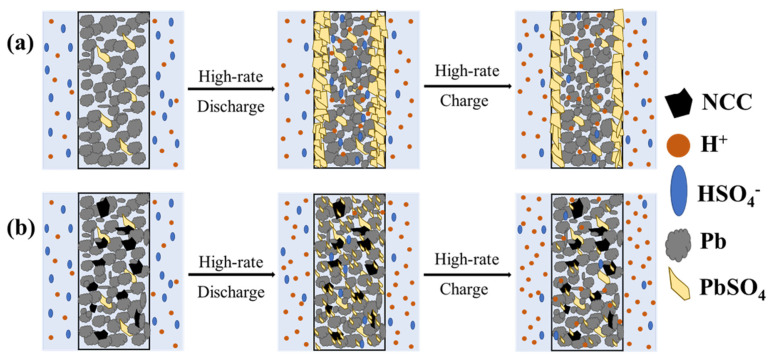
Schematic diagram of action mechanism of NCC in negative electrode plate; (**a**) blank; (**b**) NCC battery.

**Table 1 molecules-28-05618-t001:** Simulated results of EIS.

Batteries	R_s_	Q_f_	n1	R_f_	Q_dl_	n2	R_ct_
	Ω	Ω^−1^ s^n^	−	Ω	Ω^−1^ s^n^	−	Ω
Blank	1.119	8.031 × 10^−4^	0.6404	6.030	5.695 × 10^−2^	0.3786	91.00
0.5 wt.% NCC	0.7723	3.227 × 10^−2^	0.7118	0.3357	42.24 × 10^−2^	0.4950	7.224
1 wt.% NCC	0.4793	4.182 × 10^−3^	0.5504	0.6868	25.16 × 10^−2^	0.5677	2.954
1.5 wt.% NCC	0.9779	2.740 × 10^−3^	0.6280	2.904	7.173 × 10^−2^	0.5242	10.63
3 wt.% NCC	1.034	1.271 × 10^−3^	0.7626	0.9185	2.911 × 10^−2^	0.4490	17.48
1 wt.% AC	0.9764	7.499 × 10^−4^	0.7207	2.180	3.239 × 10^−2^	0.5997	122.6

**Table 2 molecules-28-05618-t002:** Cycle life of batteries with different additives.

Battery	Cycle Numbers
Blank	809
0.5 wt.% NCC	1180
1 wt.% NCC	4324
1.5 wt.% NCC	2961
3 wt.% NCC	1688
1 wt.% AC	1834

## Data Availability

The data presented in this study are available in the article.

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
