# Peer review of "Preparation of NH_4_Cl-Modified Carbon Materials via High-Temperature Calcination and Their Application in the Negative Electrode of Lead-Carbon Batteries"

_molecules, 2023, doi:10.3390/molecules28145618_

Round 1

Reviewer 1 Report

This manuscript describes a method for preparing carbon as negative electrode additive for lead-acid batteries. The idea of Chitosan-derived carbon materials/additives has been around and several researchers have conducted research upon it. This paper provides a detailed investigation of their derived NCC material. Overall, the characterization and explanation/discussion seem comprehensive. However, several issues will need to be addressed before it can be published.

Major comments:
1. Check grammar of the manuscript. There are several typos and grammar mistakes found in the text. For example, no capitalized word after comma (line 53), wrong grammar (line 73, when used).

2. The authors never gave an explanation of the acronym of NCC, which I assume is a type of carbon, please specify.

3. In Figure 5, there are several smoothed-out colored curves, which seem to be not mentioned in the text, please explain those either in caption or in text.

4. It's interesting to see in Figure 7 how 1 wt% is the turning point for both cycle life and discharge capacity, more discussion and elaboration on this part would be very helpful.

5. In Figure 8 a, 0.5 and 1 wt% NCC XRD seemed pretty clean, while 1.5 and 3 wt% NCC are noisier with PbSO4. Is this because of the activation chemistry? Please explain.

6. Please specify in Figure 9 which are before and which are after; Also, they are in different magnification, so they are not comparable.

Typo check and grammar check needed

Author Response

Please see the attachment。

Reviewer 2 Report

This manuscript  reported a modified carbon materials with NH4Cl via high temperature annealing and evaluated its addition to enhance the electrode properties. The material is interesting and the mechanism for blocking PbSo4 accumulation and surface sulfation is promising. There are several points need to be addressed:

1) what's the full name of NCC? should provide at the very beginning.

2) Where is the AC obtained and what's the AC properties, especially the composition and morphology? Why was the 1% AC chosen as the reference rather than other ratio?

3) did the AC contain nitrogen and if so, how much of N content in the AC? It seems that N atoms play a key role for the electrochemical performance as well.

4) Based on the results, NCC performs a better discharge and charge behavior than AC, but the author should discuss the reason why it can be in details. Is it because of the porous structure or the composition, morphology??

5) SEM-EDX is not a accurate tool to determine its composition quantitatively. Other analytical method should be provided.

6) What's the scale bar of edx mapping images in Figure 2C and which area are used for EDX?

7) which curve was used to determine the pore size distribution, the adsorption or desorption curve? Should state it clearly in Figure 4.

 8) Page 7, line 255-259: It seems that the value of the 3wt% is the smallest. Can author explain why?
